# Effects of different marination conditions on the physico-chemical and microbiological quality of anchovy (*Engraulis encrasicolus*) fillets inoculated *with Morganella psychrotolerans* during cold storage

**Oluwatosin Abidemi Ogunkalu**●*, Ilknur Ucak

Faculty of Agricultural Sciences and Technologies, Nigde Omer Halisdemir University, Nigde, Turkey

* ogunkaluoluwatosin1@gmail.com

**Data Availability Statement:** All relevant data are within the manuscript. A file containing Data set of value of data behind mean and standard deviation has been uploaded as supporting information.

## Abstract

This study is aimed to determine the effects of different marination conditions (1, 2, 3, 4% acetic and 6, 8, 10% NaCl) on the anchovy fillets inoculated with *Morganella psychrotolerans* during refrigerated storage (4±1˚C) for three months. According to the results of study, marination has great inhibitory effects on the growth of *M. psychrotolerans*. Total psychrophilic bacteria, total lactic acid bacteria, total yeast and mold, Total Enterobacteriaceae and *M. psychrotolerans* growth were not observed in the groups treated with 3 and 4% acetic acid. Control groups and fillets marinated with 1% acetic acid showed lower sensory scores. Those groups were rejected on 30th, 45th and 60th days of the storage, respectively, while the groups marinated with 2%, 3%, and 4% acetic acid had higher sensory scores and they were still acceptable until at the end of the study. According to peroxide value (PV) and thiobarbituric acid reactive substances (TBARS) assessment, lipid oxidation was delayed in the fillets marinated with high acetic acid concentrations (3 and 4%) comparing with the control and other inoculated fillets. From this research it can be revealed that high acetic acid and salt concentrations suppress the bacteria growth in the anchovy fillets. Thus, marination process can be recommended to be used as a preservation method to inhibit bacterial growth in anchovy fillets for a safe consumption.

## Introduction

Marination is a traditional food processing technique which improves the flavor, textural and structural properties of raw material and prolong the shelf life of product. Marinating seafood products involves processing them with acids, sodium chloride, glucose, spices, and oil to enhance the taste and texture of fisheries' products [1,2]. It likewise allows modification of the structure and tenderized texture alteration by making the fish sample last longer [3–5]. Marination is a technological procedure that does not involve heat treatment and is based on the degree of ripening; the quality of the final product is also determined by the raw material used.

**Funding:** This study was supported by the Scientific Research Projects Unit of Nigde Omer Halisdemir University with TGT 2021/19-LÜTEP project number. Also this study is produced from the PhD thesis of Oluwatosin Abidemi OGUNKALU.

**Competing interests:** The authors have declared that no competing interests exists.

Additionally, the fish material commonly determines marinating solution maturation: brining solution and sodium chloride: the ratio of acetic acids and temperature [6].

Histamine intoxication from seafood is a harmful food disease resulting from ingesting an increased quantity of histamine [7]. The aggregation of histamine in food occurs through histamine-forming bacteria that form histidine decarboxylase (HDC), an enzyme that breaks down free histidine and converts it to histamine [8]. When histamine occurs in food, it is impossible to eradicate it through general food processing techniques like heat and freezer storage [7]. In general, mesophilic bacteria like *Morganella morganii*, *Photobacterium damselae*, and *Klebsiella pnuemoniae* have a lot of records of highly histamine-forming bacteria in fish and seafood products. This mesophilic histamine-forming would hinder the formation of histamine through the control of the temperature of storage during processing [9], although psychrotrophic histamine-producers like a few Photobacterium spp. are competent to form histamine to a damaging level in fisheries products kept at refrigerator temperature, and the conventional cold chain cannot eliminate histamine formation. *Morganella psychrotolerans* is an innovative and powerful psychrotolerant histamine-producer bacteria, which can grow at a temperature between 0–2˚C. Its biological and chemical properties are closely related to *M. morganii* [10]. Current research in Japan and Denmark revealed psychrotolerant histamine poisoning bacteria (HPB). *Morganella psychrotolerans* results in more histamine fish poisoning (HFP) than the common familiar mesophilic HPB [11]. There is a harmful aggregation of histamine at 0–5˚C from the psychrotolerant extracted, and they are presented to be crucial in histamine development in chilled aquatic foods. Since histamine formation by *M. psychrotolerans* could not be hindered through only chilling fisheries products between 0–5˚C, including additives that could prevent spoilage is needed [11]. Several researchers have conducted various studies on fish-marinated product's shelf life and their attributes [4,12–14], and fisheries product marinate with the use of additives such as basil, pepper, garlic [3], rosemary extraction, yogurt, tomato blend, and other spices. Recently, several additives have been incorporated into human food: artificial preservation, coloring, and anti-oxidizing agents applied for a prolonged duration of food [3]. While Ucak et al. [15] conducted a study on the inhibitory effects of high-pressure treatment on microbial growth and biogenic amine formation in marinated herring inoculated with *M. psychrotolerans*, however, no work has been carried out on the effects of marination with a combination of acetic acid and salt on anchovy inoculated with *M. psychrotolerans* in chilling between 0–5˚C.

Thus, the objective of present study was to determine the effects of different marination conditions (1, 2, 3, 4% acetic and 6, 8, and 10% NaCl) on the anchovy fillets inoculated with *M. psychrotolerans*. In order to assess the effectiveness of the different acetic acid and NaCl concentrations pH, lipid oxidation, microbial growth and sensory changes were studied on anchovy fillets days during storage under refrigeration (4±1˚C) conditions.

## Material and methods

### Fish samples

Anchovy fillets were bought from a local market in Nigde and transported in ice to the laboratory of Animal Production and Technologies Department, Faculty of Agricultural Sciences and Technologies. Fresh anchovy fillets were kept inside polythene bags and were stored at -80˚C until marination process.

### Bacterial strain

*Morganella psychrotolerans* (DSM, 1786) was procured from the German Collection of Microorganisms and Cell Culture (DSMZ, Braunschweig, Germany). The bacterium was cultivated

through the medium of Trypticase soy Broth (TSB, Oxoid) at 25˚C overnight. The procedure was repeated by suspending bacteria cells in the same medium at $10^6$. The fish-to-bacterium medium concentration for the inoculating method was 100g/mL [15].

## Inoculation and marination of the samples

Frozen anchovy fillets were allowed to thaw overnight at 4±1˚C. Anchovy fillets were submerged in the bacterial cultured medium for 5 minutes to inoculate bacterial cells on the exterior part of the fish muscles. The fish-to-bacterium medium concentration for the inoculation procedure was 100/1, g/mL [15]. Different marination solutions were prepared with 1, 2, 3, and 4% acetic acid (v/v) and 6, 8 and 10% NaCl (w/v). The groups marinated with 1% acetic acid and 6, 8 and 10% NaCl were coded as GR1, GR2 and GR3; the groups marinated with 2% acetic acid and 6, 8 and 10% NaCl were coded as GR4, GR5 and GR6; the groups marinated with 3% acetic acid and 6, 8 and 10% NaCl were coded as GR7, GR8 and GR9; the groups marinated with 4% acetic acid and 6, 8 and 10% NaCl were coded as GR10, GR11 and GR12. Additionally, one group was left as control (C) and the other group left without marination treatment after inoculation of bacteria (CM). The fish-to-solution ratio was 1:1.5(w/v). The marination procedure was carried out according to the method of Ucak et al. [15] at 4±1˚C for 72 hours. After the marination process was concluded, the fish fillets were withdrawn from the mixture and suspended for dripping of the water on a sterilized iron strainer for 30 minutes (see Fig 1).

## Physico-chemical analysis

The assessment of pH value was conducted with the use of a pH-meter probe of the (Thermo Scientific Orion 2-star, Germany) the probe was inserted inside the fish homogenates produced with sterilized water (1:1, w:v, fish: distilled water) [16]. The peroxide value (PV) was evaluated based on AOAC techniques [17]. 2 grams of fish sample were mixed with 30 mL of a solution consisting of glacial acetic acid: chloroform (2:3, v/v). After that, 1 mL of saturated potassium iodide solution was added and the mixture was put in a dark place for 5 min. The mixture was taken and 75 mL of distilled water were added, then the mixture was titrated with 0.1 M sodium thiosulfate with the addition of a starch solution as an indicator. The results were calculated as meq $O_2$/kg. Thiobarbituric acid reactive substances (TBARS) analysis was assessed based on the methods of Tarladgis et al. [18]. Fish oil solved in n-buthanol and it was mixed with the same amount of TBA reagent (0.288 g/100 mL). The absorbance of the samples was measured with an UV-VIS spectrophotometer at 530 nm after incubation at 95˚C for 120 min in a water bath for the color reactions. Results were expressed as:

TBARS (mg malondialdehyde (MDA)/kg) = 50 x (Absorbance of lipid-Absorbance of blank)/sample weight (mg)

## Microbiological analysis

To evaluate microbial analysis, (10 g) anchovy produced for sampling was blended with 90 mL of a prechilled sterilized ringer s. Additional decimal serial dilutions were then used from the homogenate. For the isolation of total psychrophilic bacteria and total viable counts, Plate Count Agar (PCA) Oxoid was applied for its isolation. Then, the plates were incubated at 8˚C for seven days and 37˚C for 24–48 h, respectively. Yeast and mold were enumerated by plating on Potato Dextrose Agar (PDA, with pH 3.5 by addition of citric acid) and incubated at 25˚C for five days. To determine Enterobacteriaceae, the pour plating method was employed in Violet Red Bile Agar (VRBA), and the plates were incubated at 37˚C for 36–48 h. *M. psychrotolerans* was isolated through the spread of the mixed samples on Tryptic Soy Agar plates (Oxoid),

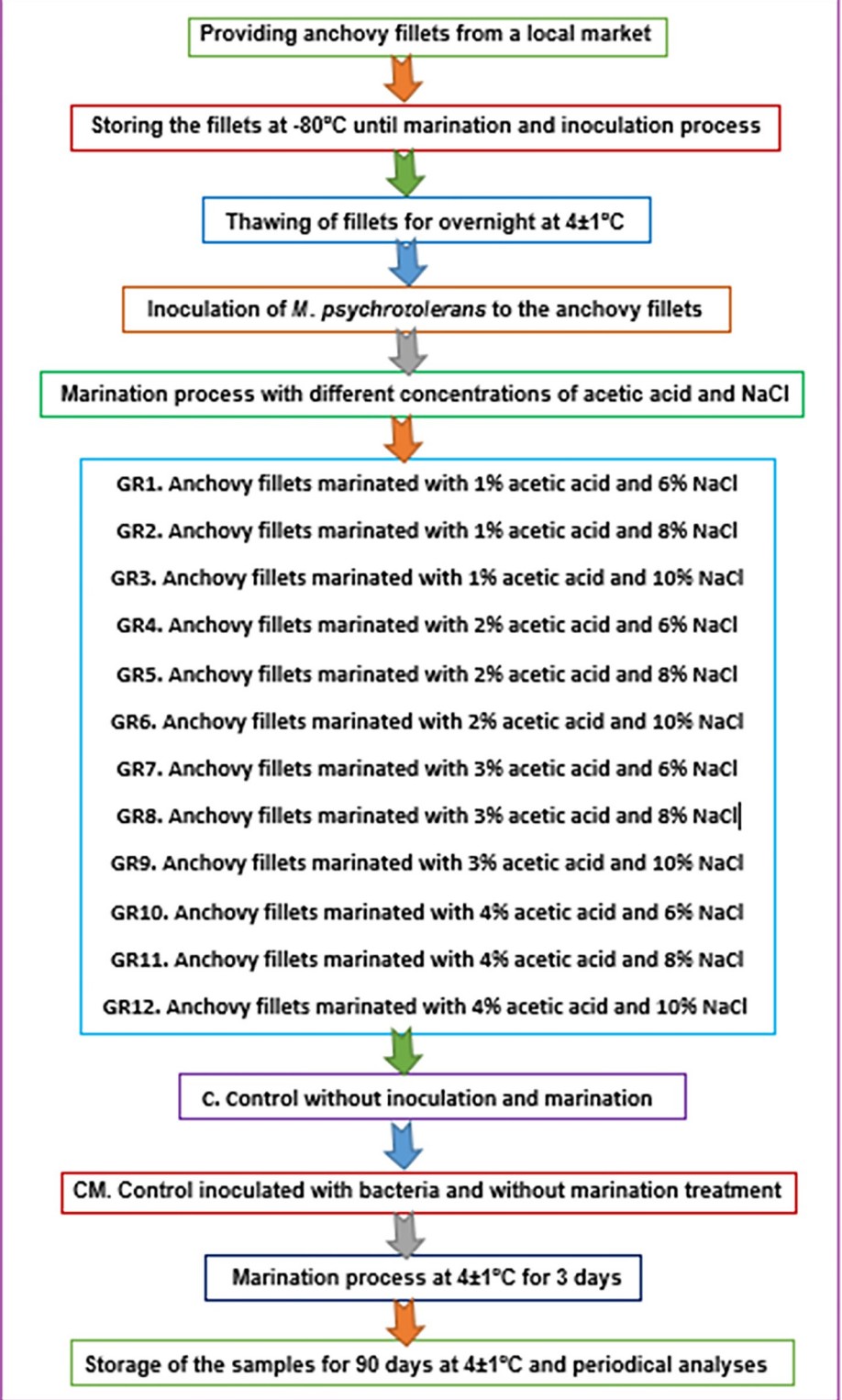

**Fig 1. The process of sample preparation (Adapted by Ilknur Ucak).**

and incubation was carried out at 25°C for 48h. Lactic Acid Bacteria were isolated through the method of pour plating 1 mL of sample on MRS agar (Merck, Darstadt, Germany). Petri dishes were incubated at 30°C for 48h. When the incubation process was completed, the colonies were counted and recorded as log CFU/g.

## Sensory analysis

The sensory assessment was conducted based on the method of Amerina et al. [19]. Eight panelists aged between 35 and 45 years who have fish consumption habit participated in sensory test. The samples were blind-coded with 3-digit random numbers. Separated sensory test boxes were used for the evaluation of samples under daylight at 24°C. The panelists assessed the sensory characteristics of fish samples based on their odor, texture, color, appearance, and general acceptance by applying a nine-point hedonic scale. A score of 9–7 means "very good," a score of 6.9–4.0 good, and a score of 3.9–1.0 was indicated as spoiled and was not accepted.

## Statistical analysis

All measurements were conducted in triplicate. Data were subjected to Analysis of Variance (ANOVA) and Duncan's multiple range tests using the SPSS Version 18.0 statistical package (SPSS Inc., Chicago, IL, USA). Differences were regarded statistically significant at $p < 0.05$.

## Results and discussion

### Effects of different marination conditions on the *Morganella psychrotolerans* growth

The effects of different marination conditions on the growth of *M. psychrotolerans* in anchovy fillets are presented in Table 1. Bacterial growth was observed in control samples (C and CM) till 15th day of the storage and in the samples marinated with 1 and 2% acetic acid and different concentrations of NaCl (6, 8, 10%). *Morganella psychrotolerans* growth reached 3.48, 4.48 and

**Table 1. The effects of different marination conditions on the growth of *M. psychrotolerans* (log CFU/g).**

| | Storage period (days) | C | CM | GR1 | GR2 | GR3 | GR4 | GR5 | GR6 | GR7 | GR8 | GR9 | GR10 | GR11 | GR12 |
|---|---|---|---|---|---|---|---|---|---|---|---|---|---|---|---|
| *M. psychrotolerans* | 0 | - | - | - | - | - | - | - | - | - | - | - | - | - | - |
| | 3 | 2.30 ±0.00^Ab | 2.45 ±0.01^Ab | 2.04 ±0.00^Ac | 2.24 ±0.02^Be | 1.88 ±0.01^Ce | 1.61 ±0.01^Dc | 1.61 ±0.02^CDc | - | - | - | - | - | - | - |
| | 15 | 2.46 ±0.01^Ba | 2.48 ±0.00^Aa | 2.36 ±0.03^Cb | 2.29 ±0.01^Dd | 2.09 ±0.01^DEd | 0.74 ±1.04^Ec | 1.83 ±0.07^Ec | 1.57 ±0.04^Fa | - | 1.56 ±0.02^Fa | 0.50 ±0.71^Fa | - | - | - |
| | 30 | - | - | 2.48 ±0.00^Aa | 2.43 ±0.00^Bc | 2.33 ±0.02^Cc | - | - | 1.68 ±0.03^Fa | - | 1.83 ±0.07^Fa | 1.00 ±0.00^Fa | - | - | - |
| | 45 | - | - | 3.48 ±0.00^Aa | 3.46 ±0.00^Ab | 3.44 ±0.01^Ab | - | 0.30 ±0.43^Bc | - | - | - | - | - | - | - |
| | 60 | - | - | - | 4.48 ±0.00^Aa | 4.50 ±0.00^Ba | 2.38 ±0.03^Cab | 2.46 ±0.00^Dbc | - | - | 2.00 ±0.00^Ea | - | - | - | - |
| | 75 | - | - | - | - | - | 3.20 ±0.28^Aab | - | - | - | 1.00 ±0.90^Ba | - | - | - | - |
| | 90 | - | - | - | - | - | 3.33 ±0.21^Aa | - | 1.00 ±0.70^Ba | - | 1.20 ±0.85^Ba | - | - | - | - |

Statistical differences were marked with different letters. Capital letters indicate significant difference among groups and lower-case letters indicate significant difference among storage days. '-' means growth not found.

4.50 CFU/g in the GR1, GR2 and GR3 groups on 45th and 60th days, respectively. Until these days bacteria growth consistently increased in the groups treated with 1% acetic acid and 6, 8 and 10% NaCl concentration. At the end of the storage period *M. psychrotolerans* growth was observed in GR4, GR6 and GR8 groups as 3.33, 1.00 and 1.20 log CFU/g, respectively. Although *M. psychrotolerans* growth was observed until at the end of the storage period in these groups, viable cell counts of *M. psychrotolerans* were significantly ($p < 0.05$) lower than those of the groups treated with 1% acetic acid and 6, 8 and 10% NaCl concentrations. Bacteria growth was detected in GR9 on the 15th and 30 days of the storage as 0.50 and 1.00 log CFU/g, respectively. The most effective acetic acid concentration was 4% incorporated with 6, 8 and 10% NaCl concentrations for the inhibition of bacteria growth. Combination of high acid concentration and salt concentration significantly delayed the bacterial growth since there were no bacteria cells in samples prepared with 4% acetic acid.

Our findings are similar to the report of Ucak et al. [15]. Their result indicated that marinate solution, which consists of 4% acetic acid and salt solution, could hinder the division of bacterial cells of *M. psychrotolerans* in herring fillets by preventing bacterial propagation. The result showed that a high concentration of acetic acid, like 3% and 4%, combined with a high concentration of salt, could prevent and hinder the multiplication of *M. psychrotolerans*, as there was no growth in those groups during the study period. *Morganella psychrotolerans* has been identified to be among the dominant spoiling microflora extracted from lumpfish roe kept at a temperature of 5°C (pH 5.4 and 3.5–4.8% sodium chloride in $H_2O$ phase) [10]. Also, *M. psychrotolerans* has been found to cause the production of dangerous concentrates of histamine found in unprocessed vacuum-packed tuna and vacuum-packed cold-smoked tuna, which resulted in HFP occurrence [10,20]. Marination is a semi-preservation technique and the preserving principle is the combination of acetic acid and salt [21]. The effects of different marination conditions on the microorganisms have been widely studied. However its effectiveness against *M. psychrotolerans* has not been studied. There some other studies which determined the suppressing effects of different additives on the growth of *M. psychrotolerans*. Wang et al. [22] studied the antimicrobial effectiveness of oregano essential oil against *M. psychrotolerans* and they reported that oregano essential oil exhibited a significant inhibitory effect on *M. psychrotolerans* in tuna. In another study it was determined that the aqueous extract of propolis more effective than ethanolic extract to decrease the growth of *M. psychrotolerans* [23]. Our observation pointed out that the combination of 4% acetic acid and NaCl had much more suppressing effect on the growth of *M. psychrotolerans*, since during the storage period bacteria cells could not grow in those groups.

## Effects of different marination conditions on the microbiological quality of anchovy

The growth of total psychrophilic and total mesophilic bacteria in marinated anchovy are presented in Table 2. The initial value of total psychrophilic bacteria was detected as 1.48 log CFU/g and total psychrophilic bacteria count showed an increase in the controls (C and CM) and in the groups marinated with 1 and 2% acetic acid during storage period. Total psychrophilic bacteria count was determined as 4.44, 5.48 and 5.42 log CFU/g in GR1, GR2 and GR3 on 45th and 60th days of the storage, respectively. Total psychrophilic bacteria count of C and CM reported as 4.19 and 4.78 log CFU/g on the 15th day of storage. This value reached 4.06 and 4.47 log CFU/g in GR4 and GR5 groups at the end of the storage, while the was no growth in the group GR6 and the groups marinated with 3 and 4% acetic acid. This may be the reason of high acidity which suppress the growth of bacteria and cause the bacteria cell damage.

Total aerobic mesophilic bacteria growth was not observed at the beginning of the storage. In the control groups and in the marinated groups except for GR6, GR7, GR8, GR9 and GR10

**Table 2. The effects of different marination conditions on the total psychrophilic and total mesophilic bacteria counts (log CFU/g).**

| | Storage period (days) | C | CM | GR1 | GR2 | GR3 | GR4 | GR5 | GR6 | GR7 | GR8 | GR9 | GR10 | GR11 | GR12 |
|---|---|---|---|---|---|---|---|---|---|---|---|---|---|---|---|
| **Total psychrophilic bacteria** | 0 | 1.48 ±0.00^Ac | 1.48 ±0.00^Ab | 1.48 ±0.00^Ac | 1.48 ±0.00^Af | 1.48 ±0.00^Af | 1.48 ±0.00^Ac | 1.48 ±0.00^Ad | 1.48 ±0.00^Ab | 1.48 ±0.00^Aa | 1.48 ±0.00^Aa | 1.48 ±0.00^Aa | 1.48 ±0.00^Aa | 1.48 ±0.00^Aa | 1.48 ±0.00^Aa |
| | 3 | 2.46 ±0.00^Ab | 2.48 ±0.00^Ab | 2.41 ±0.03^Ab | 2.28 ±0.02^Be | 1.75 ±0.08^Ce | 1.57 ±0.14^Dc | 1.67 ±0.06^CDd | - | - | - | - | - | - | - |
| | 15 | 4.19 ±0.06^Ba | 4.78 ±0.72^Aa | 3.29 ±0.02^Cb | 2.48 ±0.00^Dd | 2.27 ±0.02^DEd | 1.83 ±0.03^Ec | 1.85 ±0.00^Ed | - | - | - | - | - | - | - |
| | 30 | - | - | 437 ±0.04^Aa | 3.48 ±0.00^Bc | 2.48 ±0.00^Cc | 2.29 ±0.02^Dbc | 2.11 ±0.16^Ebc | - | - | - | - | - | - | - |
| | 45 | - | - | 4.44 ±0.05^Aa | 4.35 ±0.07^Ab | 3.47 ±0.01^Ab | 1.24 ±1.75^Bc | 1.15 ±1.63^Bd | - | - | - | - | - | - | - |
| | 60 | - | - | - | 5.48 ±0.00^Aa | 5.42 ±0.00^Ba | 2.97 ±0.72^Cab | 2.48 ±0.00^Dbc | - | - | - | - | - | - | - |
| | 75 | - | - | - | - | - | 5.53 ±0.10^Aab | 3.48 ±0.00^Bab | - | - | - | - | - | - | - |
| | 90 | - | - | - | - | - | 4.06 ±0.62^Aa | 4.47 ±0.01^Aa | - | - | - | - | - | - | - |
| **Total mesophilic bacteria** | 0 | - | - | - | - | - | - | - | - | - | - | - | - | - | - |
| | 3 | 2.45 ±0.04^Ab | 2.47 ±0.00^Ab | 2.14 ±0.09^Ad | 2.08 ±0.01^ABe | 2.07 ±0.03^ABe | 1.45 ±0.21^Be | 0.74 ±1.04^Ccd | - | - | - | - | - | - | - |
| | 15 | 4.32 ±0.03 Aa | 4.43 ±0.01^Aa | 3.21 ±0.03^Bc | 3.14 ±0.04^Bd | 2.48 ±0.00^Cd | 1.57 ±0.04^De | 0.83 ±1.17^Ecd | - | - | - | - | - | - | - |
| | 30 | - | - | 5.27 ±0.02 Ab | 4.22 ±0.02^Bc | 3.35 ±0.04^Cc | 2.27± 0.03^Dd | 1.83 ±0.18^Ec | - | - | - | - | - | - | 0.50 ±0.71^Fb |
| | 45 | - | - | 5.48 ±0.00^Aa | 4.45 ±0.00^Bb | 4.42 ±0.01^Bb | 3.48 ±0.00^Cc | 1.97 ±0.10^Cc | - | - | - | - | - | 2.33 ±0.01^Dc | 1.84 ±0.27^Ea |
| | 60 | - | - | - | 5.32 ±0.03^Aa | 5.29± 0.02^Aa | 4.47± 0.01^Bb | 3.48 ±0.00^Cb | - | - | - | - | - | 2.33 ±0.01^Dc | 1.84 ±0.27^Ea |
| | 75 | - | - | - | - | - | 5.42 ±0.00^Aa | 4.48 ±0.00^Bab | - | - | - | - | - | 2.43 ±0.02^Eb | 2.07 ±0.09^Da |
| | 90 | - | - | - | - | - | 5.70 ±0.72^Aa | 5.21 ±0.02^Aa | 1.48 ±0.00^Ca | - | - | - | - | 2.46 ±0.02^Ba | 2.15± 0.06^Ba |

Statistical differences were marked with different letters. Capital letters indicate significant difference among groups and lower-case letters indicate significant difference among storage days. '-' means growth not found.

this value showed increase. At the end of the storage, the lowest total mesophilic bacteria count was determined in GR11 and GR12 as 2.45 and 2.15 log CFU/g among the marinated fillets, while the highest values were observed in GR4 and GR5 as 5.70 and 5.21 log CFU/g, respectively. With the increasing acidity of anchovy fillets, it can be seen that the inhibitory effects on the bacteria growth increase. This value was reported as 5.48, 5.32 and 5.29 log CFU/g in the groups treated with 1% acetic acid on 45th and 60 days of the storage, respectively.

According to Sen and Temelli's [24] research on the microbial and chemical qualities of anchovies processed with marinate, the observed total aerobic mesophilic bacteria (TAMB) value in anchovy marinades was 2.6 log CFU/g. The maximum acceptable limit for the TAMB value is 6 log CFU/g ICMSF [25]. In our study, the TAMB value does not surpass the regulated limit, as shown in Table 2. According to the report of Ucak et al. [15], their findings reported no bacterial cells were seen for the groups treated with 4% acetic acid; this report agrees with the results of our study. Also, in the research of Fuseli et al. [26], they reported there was no

isolation of psychrophilic bacteria in marinated anchovy (*Engraulis anchoita*). Similarly, the findings of Cosansu et al. [27] are similar to our work as they reported increased growth in the observation of marinated anchovy with the inclusion of pneumococcus culture, which does not exceed the regulated limit. Fish marination of anchovy was shown to inhibit microorganisms due to the addition of acetic acid and sodium chloride. The inhibiting reaction of acetic acid, specifically on yeast and bacteria, was recorded by Sen and Temelli's [24]. In several past research studies, microorganism isolation was not expected during processing in marinating fish; marinate has been established to prevent the growth of microorganisms [28,29].

The changes of the lactic acid bacteria (LAB) growth in marinated anchovy were presented in Table 3. At the beginning of the study, LAB of raw fish was as 1.08 log CFU/g and this value increased in control groups (C and CM) and the fillets marinated with 1% acetic acid. Total lactic acid bacteria count reached 2.48 and 3.48 log CFU/g on 15th day of the storage in C and CM groups, respectively (p<0.05). This value was reported as 3.50 log CFU/g on the 45th day of storage in the fillets treated with 1% acetic acid and 6% NaCl, while it was determined as 4.41 and 4.48 log CFU/g on the 60th day of the storage in the groups marinated with 8 and 10% NaCl, respectively. During the storage period LAB growth was not observed in the fillets marinated with 2, 3 and 4% acetic acid concentrations. Cosansu et al. [27] reported a LAB value of 2 log CFU/g isolated from anchovy marinades. The results of their study agree with the outcome of our research. Also, the findings of Kilinc and Cakli [30] recorded a lower value of LAB isolated from marinated sardine, which was 1.30 log CFU/g. Marination enhances the growth of specific LAB that is scarcely isolated in non-marinated products. Acetic acid gives an acidic environment conducive to the activity of proteolytic enzymes available in fish muscle. The produce from the proteolysis gives energy origin for the multiplication of LAB. Hence, several LAB species were isolated in marinated fish products [31].

The effects of different acetic acid and NaCl concentrations on the total yeast and mold growth in the marinated anchovy fillets were given in Table 4. At the beginning of the study, total yeast and mold growth was not observed. In control groups (C and CM) this value was determined as 2.15 and 2.30 log CFU/g on the 15th day of the storage period. In the fillets marinated with 1% acetic acid and 6% NaCl total yeast and mold counts reached 2.97 log CFU/g on the 45th days of the storage. In the fillets treated with 1% acetic acid and 8 and 10% NaCl, total yeast and mold value reached 2.99 and 2.83 log CFU/g on the 60th day of the storage, respectively. In comparison, growth was not detected in other groups marinated with 2, 3, and 4% acetic acid concentrations. The results varied significantly (p<0.05) in the yeast and mold isolated from anchovy in the control groups and the group marinated with 1% acetic acid on day 15. The research results indicate that high acetic acid combination with NaCl had inhibitory effects on microbial propagation of anchovy.

Total Enterobacteriaceae was recorded as 0.98 log CFU/g initially (Table 3). There was an increase in the growth of bacteria in the control groups (C and CM) until 15th day of the storage (2.8 and 2.42 log CFU/g). In the other groups treated with different marination solutions (1, 2, 3, and 4% acetic acid and 6, 8, and 10% NaCl) there was no growth throughout the storage. The finding of our research is like the study of Kocatepe et al. [32], which uses different essential oils in anchovy marinades and their results indicated there was no growth of total Enterobacteriaceae in marinated anchovy. Also, in the research of Fuselli et al. [29], it was reported that there was no isolation of coliform cells in cold marinades of anchovies. Likewise, the study of Gunsen et al. [21] reported there was no detection of total coliform bacteria in anchovy treated with marination stored at 4°C for eleven months. These findings agree with the present study, and it shows that marination with different concentrations of acetic acid and NaCl had inhibitory effects on the growth of Total Enterobacteriaceae.

**Table 3. The effects of different marination conditions on the LAB, total yeast and mold, total Enterobactericiea (log CFU/g).**

| | Storage period (days) | C | CM | GR1 | GR2 | GR3 | GR4 | GR5 | GR6 | GR7 | GR8 | GR9 | GR10 | GR11 | GR12 |
|---|---|---|---|---|---|---|---|---|---|---|---|---|---|---|---|
| **LAB** | 0 | 1.08 ±0.43$^{Ab}$ | 1.08 ±0.43$^{Ac}$ | 1.08 ±0.43$^{Ac}$ | 1.08 ±0.43$^{Ad}$ | 1.08 ±0.43$^{Ad}$ | 1.08 ±0.43$^{Aa}$ | 1.08 ±0.43$^{Aa}$ | 1.08 ±0.43$^{Aa}$ | 1.08 ±0.43$^{Aa}$ | 1.08 ±0.43$^{Aa}$ | 1.08 ±0.43$^{Ab}$ | 1.08 ±0.43$^{Aa}$ | 1.08 ±0.43$^{Aa}$ | 1.08 ±0.43$^{Aa}$ |
| | 3 | 1.85 ±0.11$^{Bab}$ | 2.47 ±0.01$^{Ab}$ | 0.93 ±0.04$^{Cc}$ | 1.16 ±0.65$^{Cd}$ | 0.84 ±0.34$^{Cd}$ | - | - | - | - | - | - | - | - | - |
| | 15 | 2.48 ±0.08$^{Ba}$ | 3.48 ±0.00$^{Aa}$ | 2.00 ±0.02$^{Cb}$ | 1.98 ±0.12$^{Cc}$ | 1.63 ±0.04$^{Cc}$ | - | - | - | - | - | - | - | - | - |
| | 30 | - | - | 2.47 ±0.01$^{Ab}$ | 2.47 ±0.01$^{Ac}$ | 2.12 ±0.01$^{Bbc}$ | - | - | - | - | - | - | - | - | - |
| | 45 | - | - | 3.50 ±0.00$^{Aa}$ | 3.47 ±0.01$^{Bb}$ | 2.48 ±0.00$^{Cb}$ | - | - | - | - | - | - | - | - | - |
| | 60 | - | - | - | 4.41 ±0.02$^{Ba}$ | 4.48 ±0.00$^{Aa}$ | - | - | - | - | - | - | - | - | - |
| | 75 | - | - | - | - | - | - | - | - | - | - | - | - | - | - |
| | 90 | - | - | - | - | - | - | - | - | - | - | - | - | - | - |
| **Total yeast and mold** | 0 | - | - | - | - | - | - | - | - | - | - | - | - | - | - |
| | 3 | - | - | - | - | - | - | - | - | - | - | - | - | - | - |
| | 15 | 2.15 ±0.04$^{Ba}$ | 2.30 ±0.14$^{Aa}$ | 1.66 ±0.06$^{Cc}$ | 1.57 ±0.12$^{CDc}$ | 1.46 ±0.02$^{Dd}$ | - | - | - | - | - | - | - | - | - |
| | 30 | - | - | 2.08 ±0.03$^{Bb}$ | 2.12 ±0.04$^{Ab}$ | 1.97 ±0.03$^{Cc}$ | - | - | - | - | - | - | - | - | - |
| | 45 | - | - | 2.97 ±0.02$^{Aa}$ | 2.86 ±0.02$^{Ba}$ | 2.63 ±0.04$^{Cb}$ | - | - | - | - | - | - | - | - | - |
| | 60 | - | - | - | 2.99 ±0.02$^{Aa}$ | 2.83 ±0.02$^{Ba}$ | - | - | - | - | - | - | - | - | - |
| | 75 | - | - | - | - | - | - | - | - | - | - | - | - | - | - |
| | 90 | - | - | - | - | - | - | - | - | - | - | - | - | - | - |
| **Total Enterobacteriaceae** | 0 | 0.98 ±0.03$^{Ac}$ | 0.98 ±0.03$^{Ac}$ | 0.98 ±0.03$^{Aa}$ | 0.98 ±0.03$^{Aa}$ | 0.98 ±0.03$^{Aa}$ | 0.98 ±0.03$^{Aa}$ | 0.98 ±0.03$^{Aa}$ | 0.98 ±0.03$^{Aa}$ | 0.98 ±0.03$^{Aa}$ | 0.98 ±0.03$^{Aa}$ | 0.98 ±0.03$^{Aa}$ | 0.98 ±0.03$^{Aa}$ | 0.98 ±0.03$^{Aa}$ | 0.98 ±0.03$^{Aa}$ |
| | 3 | 1.97 ±0.01$^{Bb}$ | 2.06 ±0.03$^{Ab}$ | - | - | - | - | - | - | - | - | - | - | - | - |
| | 15 | 2.38 ±0.02$^{Ba}$ | 2.42 ±0.02$^{Aa}$ | - | - | - | - | - | - | - | - | - | - | - | - |
| | 30 | - | - | - | - | - | - | - | - | - | - | - | - | - | - |
| | 45 | - | - | - | - | - | - | - | - | - | - | - | - | - | - |
| | 60 | - | - | - | - | - | - | - | - | - | - | - | - | - | - |
| | 75 | - | - | - | - | - | - | - | - | - | - | - | - | - | - |
| | 90 | - | - | - | - | - | - | - | - | - | - | - | - | - | - |

Statistical differences were marked with different letters. Capital letters indicate significant difference among groups and lower-case letters indicate significant difference among storage days. '-' means growth not found.

**Table 4. The effects of different marination conditions on the physico-chemical quality of anchovy fillets during cold storage.**

| | Storage period (days) | C | CM | GR1 | GR2 | GR3 | GR4 | GR5 | GR6 | GR7 | GR8 | GR9 | GR10 | GR11 | GR12 |
|---|---|---|---|---|---|---|---|---|---|---|---|---|---|---|---|
| TBARS (mg MAD/kg) | 0 | 0.15±0.01^Ac | 0.15±0.01^Ac | 0.15±0.01^Ae | 0.15±0.01^Af | 0.15±0.01^Ae | 0.15±0.01^Ag | 0.15±0.01^Ae | 0.15±0.01^Ad | 0.15±0.01^Ae | 0.15±0.01^Af | 0.15±0.01^Af | 0.15±0.01^Af | 0.15±0.01^Af | 0.15±0.01^Ag |
| | 3 | 1.58±0.37^Bb | 2.06±0.01^Ab | 1.41±0.03^BCd | 1.43±0.12^BCc | 1.50±0.01^BCd | 1.35±0.05^BCe | 1.35±0.09^BCe | 1.29±0.40^Cf | 1.34±0.07^BCg | 1.141±0.02^Cf | 1.21±0.02^BCb | 1.42±0.17^BCd | 1.18±0.00^BCd | 1.25±0.05^BCd |
| | 15 | 6.09±0.51^Ba | 7.18±0.00^Aa | 3.36±0.21^Cc | 1.97±0.06^Dd | 1.99±0.17^Dd | 1.82±0.09^De | 1.78±0.11^Dcd | 1.84±0.08^Dc | 1.81±0.07^Dc | 1.80±0.13^Dd | 1.69±0.13^Dde | 1.68±0.11^De | 1.76±0.03^Dd | 1.76±0.03^De |
| | 30 | - | - | 4.18±0.20^Ab | 2.63±0.11^BCc | 2.79±0.58^Bc | 2.01±0.03^Ce | 2.15±0.23^Cc | 2.01±0.04^Cb | 2.20±0.14^Cc | 2.14±0.12^Cc | 2.22±0.17^Ccd | 2.10±0.16^Cc | 2.09±0.07^Cc | 2.08±0.11^Cd |
| | 45 | - | - | 5.53±0.02^Aa | 5.38±0.16^Ab | 5.00±0.02^Bb | 2.94±0.07^Dd | 3.32±0.30^Cb | 3.38±0.23^Cb | 2.77±0.07^Db | 2.70±0.07^Db | 2.75±0.34^DEbc | 2.40±0.00^Eb | 2.48±0.04^Eb | 2.40±0.08^Eb |
| | 60 | - | - | - | 5.93±0.04^Ab | 5.93±0.08^Ab | 3.22±0.14^CDAb | 3.66±0.30^BCbc | 3.99±0.56^Bc | 3.10±0.19^CDe | 2.19±0.16^Dd | 3.10±0.60^CDab | 3.00±0.01^CDa | 2.99±0.02^CDa | 3.09±0.04^CDa |
| | 75 | - | - | - | - | - | 3.84±0.07^ABcb | 3.95±0.06^ABa | 4.32±0.40^Aa | 3.22±0.47^CDab | 2.94±0.07^Dab | 3.54±0.36^BCDa | 3.57±0.06^BCDa | 3.20±0.04^CDb | 3.32±0.12^BCDb |
| | 90 | - | - | - | - | - | 4.09±0.17^ABca | 4.23±0.47^ABa | 4.56±0.43^Aa | 3.43±0.04^CDa | 3.10±0.16^Da | 3.72±0.37^BCDa | 3.66±0.01^BCDa | 3.70±0.07^BCDa | 3.75±0.30^BCDd |
| PV (meq O₂/kg) | 0 | 1.50±0.71^Ac | 1.50±0.71^Ac | 1.50±0.71^Ae | 1.50±0.71^Af | 1.50±0.71^Ae | 1.50±0.71^Ag | 1.50±0.71^Ag | 1.50±0.71^Ah | 1.50±0.71^Ah | 1.50±0.71^Ag | 1.50±0.71^Ag | 1.50±0.71^Ad | 1.50±0.71^Af | 1.50±0.71^Af |
| | 3 | 7.00±0.00^Ab | 8.00±0.00^Ab | 5.99±0.00^Bd | 6.00±0.00^Be | 6.00±0.00^Bd | 6.00±0.00^Bf | 6.00±0.00^Bf | 6.00±0.00^Bf | 5.99±0.00^Bg | 6.00±0.00^Bf | 6.00±0.00^Bfg | 6.00±0.00^Bcd | 6.00±0.00^Be | 6.00±0.00^Be |
| | 15 | 17.00±0.00^Aa | 19.00±0.71^Aa | 7.98±0.00^Bc | 8.50±0.71^Bd | 5.00±0.00^BCd | 8.00±5.66^Bf | 7.50±0.71^BCef | 7.50±2.12^BCef | 8.98±0.00^Bf | 6.50±0.71^BCf | 3.00±2.83^Cef | 5.00±5.66^BCcd | 7.00±1.41^BCe | 6.50±2.12^BCe |
| | 30 | - | - | 18.50±0.71^Ab | 15.98±0.00^Bc | 15.50±0.71^Bc | 12.00±0.00^Ce | 10.99±0.00^Dde | 10.50±0.71^DEde | 9.98±0.00^Ee | 9.99±0.00^Ee | 10.00±0.00^Ede | 10.00±0.00^Ec | 10.50±0.71^Ed | 10.00±0.00^Ed |
| | 45 | - | - | 20.50±0.71^Aa | 20.98±1.41^Ab | 20.50±0.71^Ab | 15.50±0.71^Bd | 14.49±0.71^BCd | 13.47±0.71^BCDd | 14.96±0.00^BCd | 14.99±1.41^BCd | 14.50±0.71^BCd | 13.00±1.41^CDc | 12.00±0.00^Dd | 12.00±0.00^Dd |
| | 60 | - | - | - | 42.50±0.71^EFGa | 28.50±0.71^FGHa | 25.50±0.71^Bc | 25.95±0.00^Ac | 50.50±0.71^Cc | 26.95±1.41^Cc | 24.50±0.71^DEFc | 25.00±1.41^DEc | 40.46±0.71^EFGb | 27.50±0.71^GHc | 23.00±1.41^Hc |
| | 75 | - | - | - | - | - | 52.500±3.54^ABb | 56.00±4.26^Ab | 51.90±2.82^ABb | 49.95±0.00^BCb | 44.82±0.00^Ab | 44.00±5.66^Cb | 30.00±1.41^Da | 30.91±1.41^Db | 29.44±0.71^Db |
| | 90 | - | - | - | - | - | 60.38±0.71^Aa | 60.00±0.00^Aa | 56.50±0.71^Ba | 52.40±0.71^Ca | 47.36±0.70^Da | 49.50±0.71^Da | 33.50±21.2^Fab | 35.50±2.12^Efa | 38.35±0.70^Ea |
| pH | 0 | 6.02±0.16^Ab | 6.02±0.16^Ab | 6.02±0.16^Aa | 6.02±0.16^Aa | 6.02±0.16^Aa | 6.02±0.16^Aa | 6.02±0.16^Ab | 6.02±0.16^Aa | 6.02±0.16^Aa | 6.02±0.16^Aa | 6.02±0.16^Aa | 6.02±0.16^Aa | - | - |
| | 3 | 5.75±0.05^Ac | 5.87±0.09^Ab | 5.52±0.31^Aa | 5.18±0.59^BCc | 4.74±0.58^CDb | 4.62±0.47^DEb | 4.32±0.35^DEHd | 4.17±0.27^EFGb | 4.25±0.36^EFGHb | 4.76±1.05^DEFGb | 3.96±0.14^FGHd | 3.74±0.19^GHHc | - | - |
| | 15 | 6.86±0.06^Ba | 7.43±0.32^Aa | 4.90±1.08^Ec | 6.38±1.25^Ca | 4.58±0.41^Fc | 4.51±0.21^Fb | 4.34±0.91^Dc | 4.25±0.17^Gb | 3.81±0.39^HIIcd | 3.89±0.23^HIbc | 3.92±0.14^Hd | 3.55±0.08^Kd | - | - |
| | 30 | - | - | 5.11±2.29^Ac | 5.17±0.73^Ae | 4.70±0.53^Fd | 4.07±0.42^Hd | 4.14±0.46^Ac | 4.00±0.33^Cb | 3.71±0.17^Fcd | 3.74±0.16^Hc | 3.73±0.12^ABc | 3.52±0.09^Eb | - | - |
| | 45 | - | - | 6.17±2.76^Aa | 5.61±0.94^Dd | 5.01±0.63^Fd | 4.53±0.84^Fd | 4.16±0.75^Bb | 4.14±0.39^Eb | 3.70±0.55^Fd | 3.64±0.24^Fbc | 3.76±0.29^Cb | 3.50±0.13^Eb | - | - |
| | 60 | - | - | - | 5.03±0.71^De | 4.65±0.46^Ed | 3.90±1.74^Fd | 4.13±0.49^Ac | 4.14±0.30^Cb | 3.74±0.06^Ed | 3.71±0.22^DEbc | 3.71±0.22^Cb | 3.45±0.12^Cb | - | - |
| | 75 | - | - | - | - | - | 3.98±1.78^ABc | 4.05±1.81^Ae | 3.94±1.77^ABc | 3.73±0.12^Bcd | 3.71±0.16^CDbc | 3.83±0.08^ABCde | 3.45±0.16^Ed | - | - |
| | 90 | - | - | - | - | - | 4.07±1.82^Bc | 4.07±1.82^Ae | 4.14±0.04^ABb | 3.95±0.08^Cbc | 3.84±0.12^DEbc | 3.72±0.05^EFe | 3.61±0.17^Gcd | - | - |

Statistical differences were marked with different letters. Capital letters indicate significant difference among groups and lower-case letters indicate significant difference among storage days. '-' means growth not found.

## Effects of different marination conditions on the physico-chemical quality

Peroxide value (PV) is one of the crucial parameter which is necessary to measure primary oxidation products. At the beginning of the storage, PV of anchovy fillets was determined as 1.50 meq $O_2$/kg and this value showed increase in all groups during the storage period (Table 4). On the 15th day of the storage, the highest (p<0.05) PV was found in the control groups as 17.00 and 19.00 meq $O_2$/kg in C and CM groups, respectively, while the lowest value was reported in GR11 (3.00 meq $O_2$/kg) in the same storage day. Peroxide values of GR2 and GR3 reached to 42.50 and 28.50 meq $O_2$/kg on the 60[th] days of the storage, whereas this value was determined as 20.50 on the 45[th] day of the storage in the G1. According to the study of Gokoglu et al. [33] on marinated fish supplemented with garlic or tomato, an increased PV was observed, which agrees with the results in the current research. Likewise, the study of Ochrem et al. [34] on marinated herring with the addition of milk thistle recorded an increased trend in the measurement of peroxide value. Additionally, Topuz et al. [14] recorded a lower PV in the groups by adding olive oil-lemon (25, 35, and 50%) than in the control groups. The present results showed that low acetic acid concentration could not prevent lipid oxidation, while a high concentration of acetic acid is effective to control the delay in lipid oxidation.

Thiobarbituric acid reactive substances (TBARS) is commonly used to determine the secondary oxidation degree. The TBARS value of initial raw material was 0.15 mg MDA/kg (Table 4). This value showed increase all groups throughout the storage. In the control and CM groups TBARS value was reached to 6.09 and 7.18 mg MDA/kg on the 15[th] day. At the end of the storage period, the lowest value (p<0.05) was determined in the GR8 as 3.10 mg MDA/kg comparing with the other marinated fillets. Thiobarbituric acid reactive substances value of frozen or chilled fish products indicating good quality should be 5 mg MDA/kg of tissue [5]. In the present study, TBARS values of anchovy fillets marinated with 2, 3 and 4% acetic acid were much lower than the maximum limit during the storage period. Thiobarbituric acid reactive substances value for the group marinated with 1% acetic acid exceeded the limit value on the 45[th] and 60[th] day of the storage in GR1 and GR2, GR3, respectively. Similarly, an increase in TBARS value was recorded for marinated anchovies [21,35] and marinated rainbow trout [3] was reported in other studies. In another study, TBARS value was showed increase in marinated anchovy treated with pomegranate [36]. A similar increase was reported by Simat et al. [37] for the brining and marinated anchovy with lemon juice.

The changes in pH measurements during the storage time of marinated anchovy inoculated with *M. psychrotolerans* are presented in the Table 4. The initial value of raw material was determined as 6.02. After marination process significantly lowest pH values were observed in the groups marinated with 4, 3 and 2% acetic acid, respectively, comparing with 1% acetic acid group. It was indicated that the accumulation of alkaline compounds causes the increase of pH value [38,39]. In the present study pH remained lower due to high acetic acid concentrations and this condition supported the inhibition of microbial growth. Similarly, the reduction of pH value after the marination process was reported in different studies [3,35]. Aksu et al. [40] carried out the procedure of marinating anchovy fish through immersion in a mixture of marinate, which consists of 2% acetic acid, and the pH values they observed were 4.50 and 4.25. Erdem et al. [41] also recorded a pH measure of 4.11, and the concentration of acetic acid they applied to anchovy was 4%. Also, Gokoglu et al. [4] recorded a significantly different pH value among seafood products treated with 2% and 4% acetic acid mixtures. Also, Guiffrida et al. [42] stated a fluctuation flow in observing the pH values of marinated seabass rises after it reduced to 4.5 and increased from 4.4 to 4.9. This fluctuation in pH value is similar to the result we observed in our findings. Bilgin et al. [43] reported that the pH value obtained from marinade products typically followed a reduction pattern within the first three months of

processing and began to increase within three and five months of processing. They explained that the increased pH value after a decreased pattern could be related to the dissipation of carbon dioxide adenosine triphosphate and hydrogen ions in fish muscles, which reduces pH value. However, developing protein degradation compositions like trimethylamine and ammonia through the deterioration of bacteria results in an increment in pH value. During fish processing, lactic acid bacteria could grow, and this effect could lead to the disintegration of amino acids. Hence, the occurrence of carbon dioxide and several removals of carboxyl groups in products were detected. These chemical processes tied up the acid and pH of the marinating product [44].

### Effects of different marination conditions on the sensory quality

Sensory analysis is the standard general method by which seafood is evaluated. It takes a short period, is easy, and gives instant details of the attributes of the fish. These sensory attributes of seafood products are distinct to the public and essential for public satisfaction [45]. The sensory analysis is measurements of colors, odor, texture, appearance, and overall generally carried out by humans using their sensory receptors. When seafood products are accepted and analyzed, the characteristics are based on chemical quality, but rejected according to the sensory parameters; such seafood is not recommended to be eaten [46]. Sensory analysis is essential to establish fish durability when processed with marinate solution [30]. The groups processed with 1, 2, 3 and 4% acetic acid display higher sensory values than the control groups (C and CM) (Figs 1–5). Marination greatly impacted the qualities of anchovy texture, color,

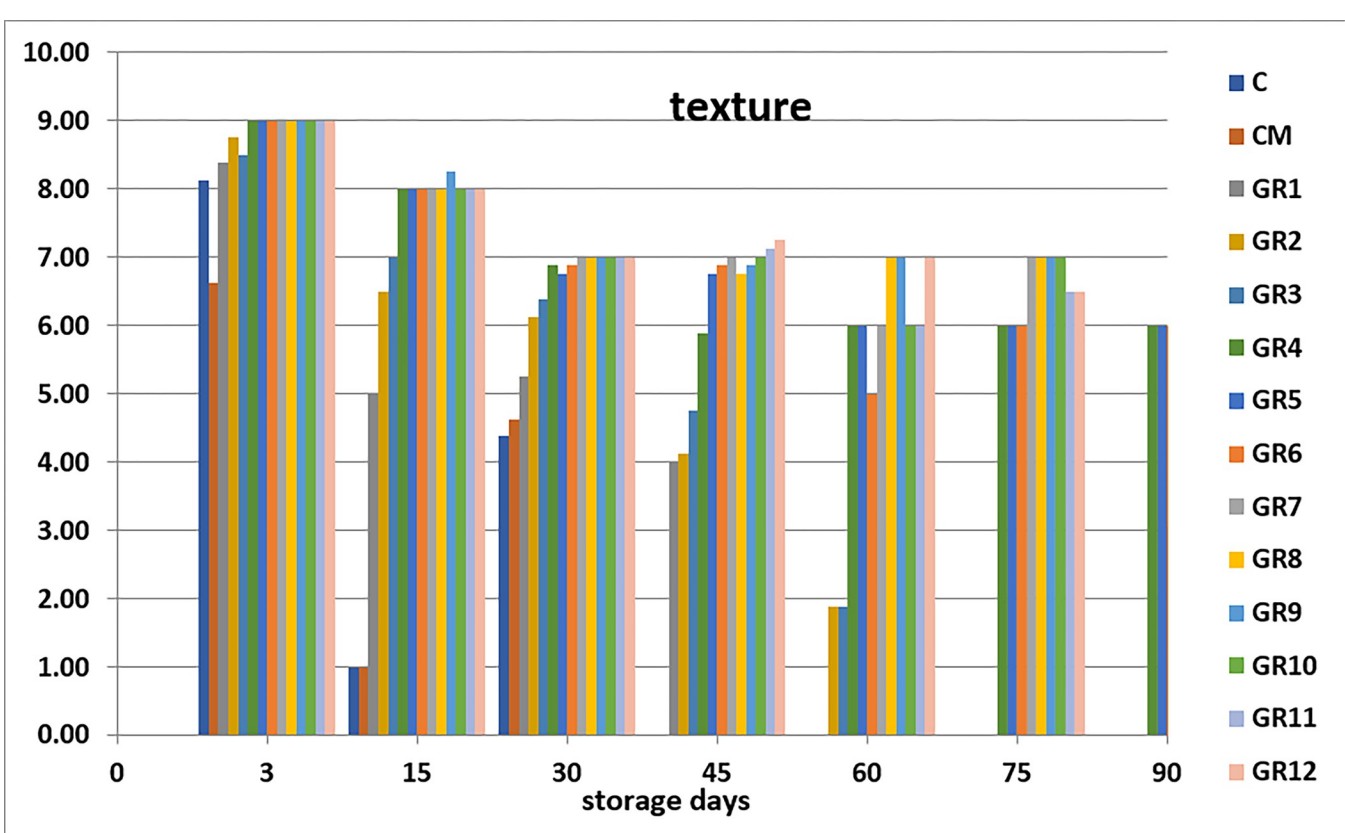

**Fig 2. Effects of different marination conditions on the texture of anchovy fillets.**

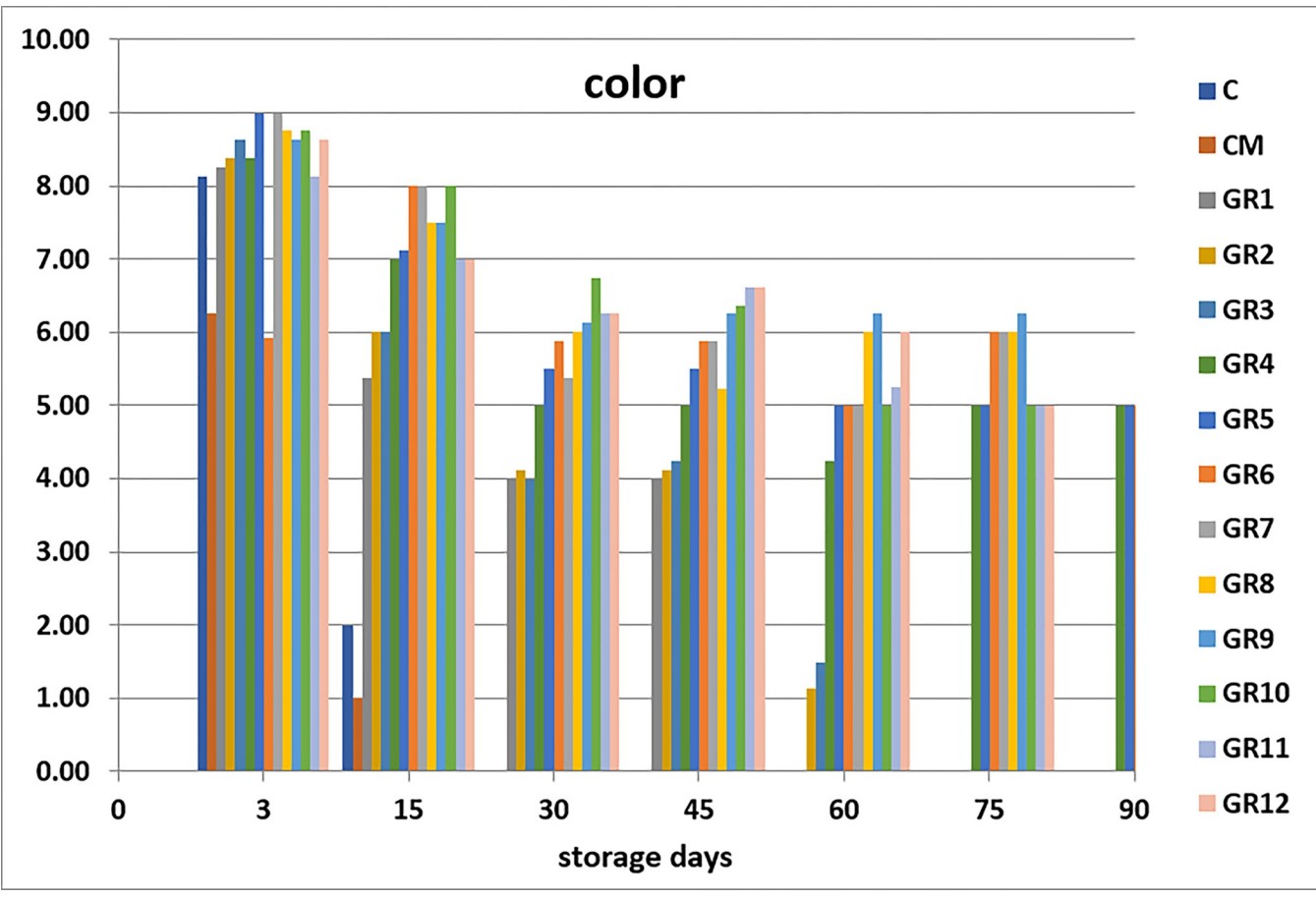

**Fig 3. Effects of differnet marination conditions on the color of anchovy fillets.**

appearance, odor, and overall quality. Hence, the groups processed with high concentration of acetic acid have the highest sensory scores. The control groups (C and CM) were rejected on day 15th of the storage because of their low scores. Meanwhile, on days 45 and 60, the panelists rejected the sample treated with 1% acetic acid. After that, the groups marinated with 2, 3 and 4% acetic acid had good sensory scores and were still acceptable until the at end of the storage period. This finding agrees with the study of Gokoglu et al. [4], which recorded marinated sardines with 2 and 4% acetic acid and preserved at 4˚C for three months to have excellent attributes. It was reported that the marination of anchovy and bonito with 10% NaCl and 4% acetic acid can extend the shelf life of product during refrigerated storage [47]. Kurt Kaya and Basturk [48] found that according to the sensory evaluation, the shelf life of marinated catfish is 110 days which treated with sunflower oil and 80 days which treated with sunflower oil and tomato sauce.

## Conclusion

Based on the research findings, it can be concluded that the usage of different concentration of acetic acid and NaCl in the marination process of anchovy fillets has preservative effects and inhibitory effects on the growth of *M. psychrotolerans*, total psychrophilic bacteria, total mesophilic bacteria, total yeast and mold and total Enterobacteriaceae. Especially 3 and 4% concentration of acetic acid incorporation with different concentration of NaCl were much more

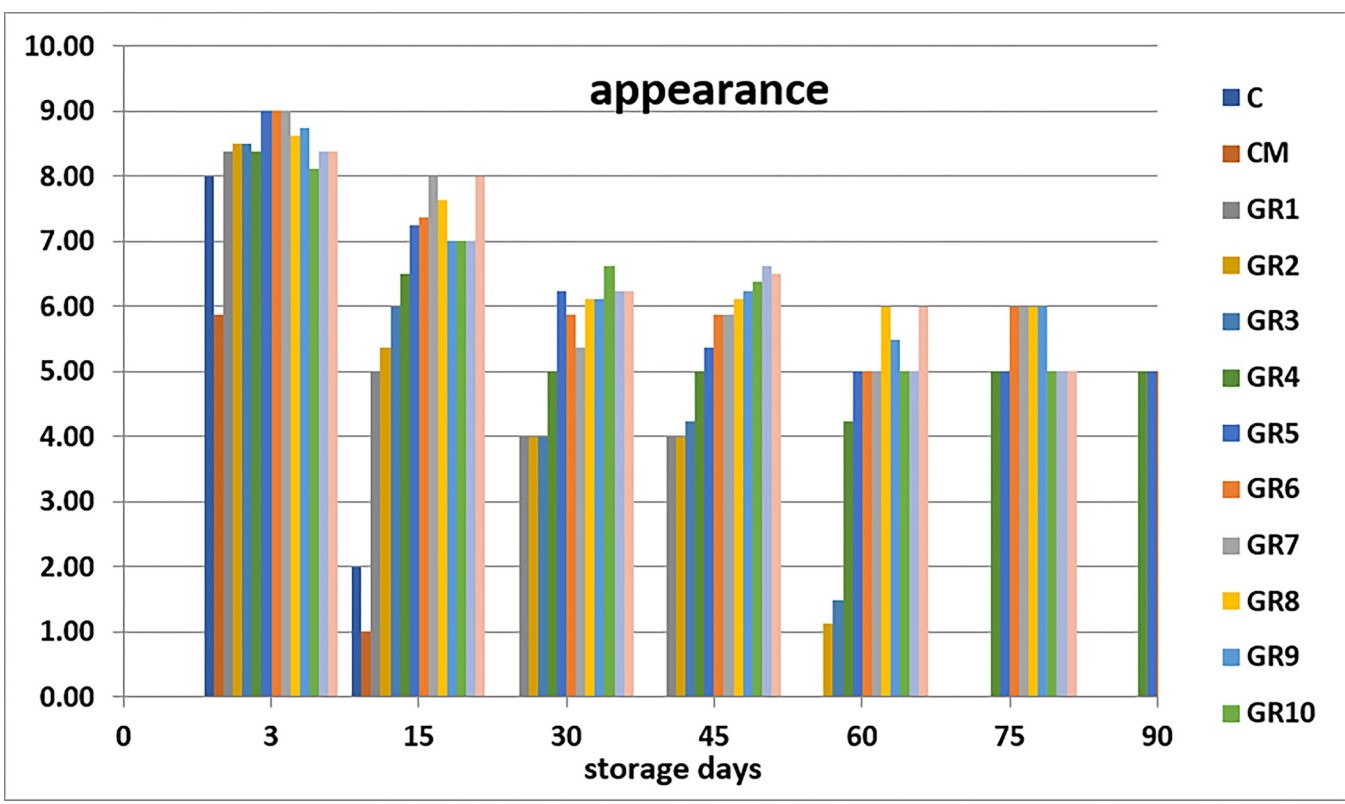

**Fig 4. Effects of different marination conditions on the appearance of anchovy fillets.**

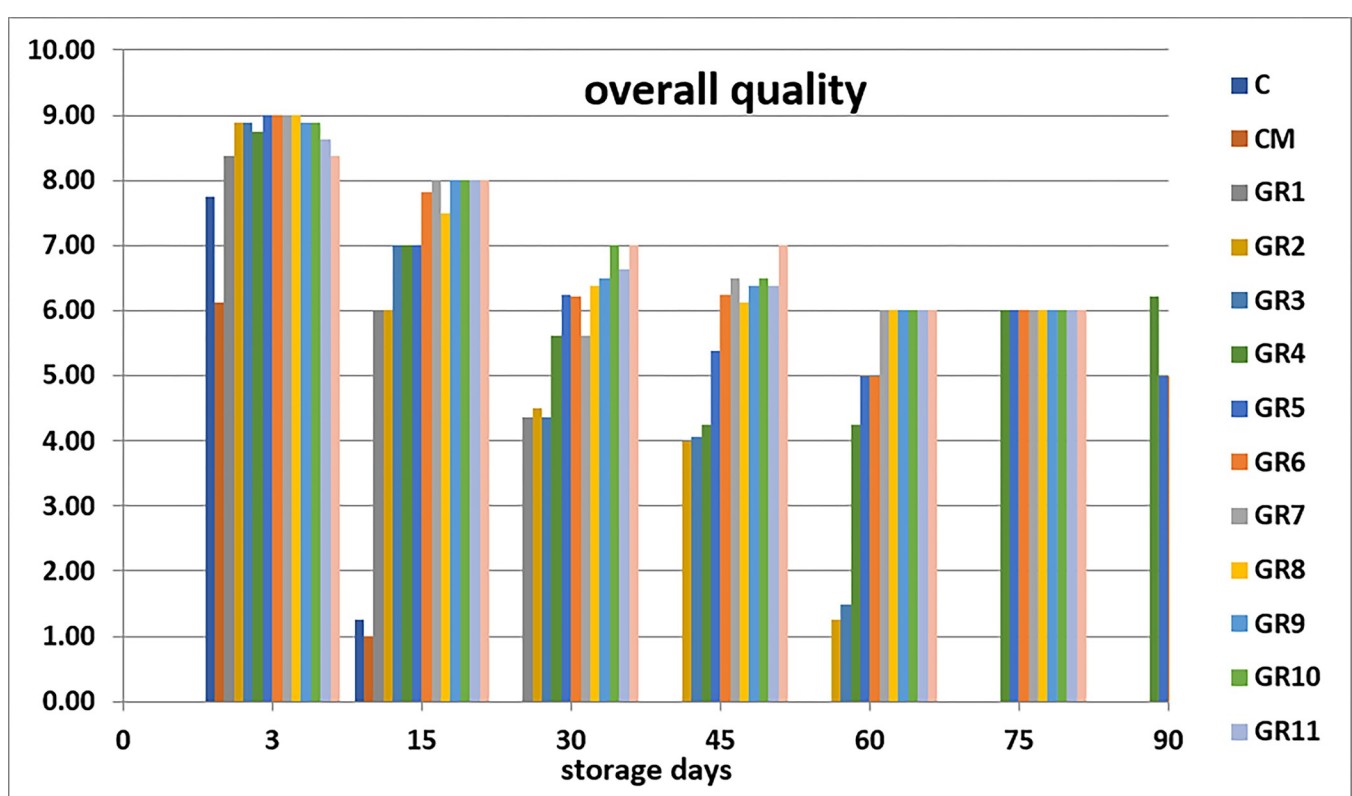

**Fig 5. Effects of different marination conditions on the overall quality of anchovy fillets.**

effective. Therefore, it can be concluded that the quality of marinated anchovy fillets can be better preserved with usage of 3 and 4% acetic acid concentrations.

## Supporting information

**S1 Data.**
(XLSX)

## Acknowledgments

I appreciate the Scientific Research Projects Unit of Nigde Omer Halisdemir University, Nigde, Turkey for their support.

## Author Contributions

**Conceptualization:** Ilknur Ucak.

**Data curation:** Oluwatosin Abidemi Ogunkalu.

**Formal analysis:** Oluwatosin Abidemi Ogunkalu.

**Funding acquisition:** Ilknur Ucak.

**Investigation:** Oluwatosin Abidemi Ogunkalu.

**Project administration:** Oluwatosin Abidemi Ogunkalu.

**Software:** Oluwatosin Abidemi Ogunkalu.

**Supervision:** Ilknur Ucak.

**Writing – original draft:** Oluwatosin Abidemi Ogunkalu.

**Writing – review & editing:** Ilknur Ucak.

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
