## [Decision Letter · Decision Letter 0]

11 Feb 2024

PONE-D-23-44219Effects of Marination conditions on the Physico-chemical and microbiological quality on anchovy (Engraulis encrasicolus) fillets inoculated with Morganella psychrotolerans during cold storage.PLOS ONE

Dear Dr. Ogunkalu,

Thank you for submitting your manuscript to PLOS ONE. After careful consideration, we feel that it has merit but does not fully meet PLOS ONE’s publication criteria as it currently stands. Therefore, we invite you to submit a revised version of the manuscript that addresses the points raised during the review process.

Please submit your revised manuscript by Mar 27 2024 11:59PM. If you will need more time than this to complete your revisions, please reply to this message or contact the journal office at plosone@plos.org. Please include the following items when submitting your revised manuscript:A rebuttal letter that responds to each point raised by the academic editor and reviewer(s). You should upload this letter as a separate file labeled 'Response to Reviewers'.A marked-up copy of your manuscript that highlights changes made to the original version. You should upload this as a separate file labeled 'Revised Manuscript with Track Changes'.An unmarked version of your revised paper without tracked changes. You should upload this as a separate file labeled 'Manuscript'.

We look forward to receiving your revised manuscript.

Kind regards,

Luca Nalbone, Ph.D.

Academic Editor

PLOS ONE

Journal Requirements:

a) The name of the colleague or the details of the professional service that edited your manuscript.

b) A copy of your manuscript showing your changes by either highlighting them or using track changes (uploaded as a *supporting information* file).

c) A clean copy of the edited manuscript (uploaded as the new *manuscript* file).

3. You indicated that ethical approval was not necessary for your study. We understand that the framework for ethical oversight requirements for studies of this type may differ depending on the setting and we would appreciate some further clarification regarding your research. Could you please provide further details on why your study is exempt from the need for approval and confirmation from your institutional review board or research ethics committee (e.g., in the form of a letter or email correspondence) that ethics review was not necessary for this study? 

Please include a copy of the correspondence as an "Other" file.

4. Please provide additional details regarding participant consent. In the ethics statement in the Methods and online submission information, please ensure that you have specified (1) whether consent was informed and (2) what type you obtained (for instance, written or verbal, and if verbal, how it was documented and witnessed). If your study included minors, state whether you obtained consent from parents or guardians. If the need for consent was waived by the ethics committee, please include this information.

5. We note that your Data Availability Statement is currently as follows: [All relevant data are within the manuscript and its supporting information files.]

**Additional Editor Comments:**

Please respond to the reviewers' comments below and edit the text accordingly. Since the reviewers highlighted the need for major revisions, a relatively significant modification of the manuscript is expected. In particular, pay attention to English. Thank you for choosing Plos One for your article.

Reviewers' comments:

Reviewer's Responses to Questions

**Comments to the Author**

1. Is the manuscript technically sound, and do the data support the conclusions?

Reviewer #1: Partly

Reviewer #2: Yes

2. Has the statistical analysis been performed appropriately and rigorously? 

Reviewer #1: N/A

Reviewer #2: No

3. Have the authors made all data underlying the findings in their manuscript fully available?

Reviewer #1: Yes

Reviewer #2: Yes

4. Is the manuscript presented in an intelligible fashion and written in standard English?

Reviewer #1: No

Reviewer #2: No

5. Review Comments to the Author

Reviewer #1: Too many spelling and grammatical errors in the text.

The title is not written with due care and the scientific names in the text are not written with due care. For example, the bacteria mentioned as "Morganella psychrotolerans" in line 28 of the abstract should be given as M. Psychrotolerans in line 32 of the same section, the names of bacteria should be italicized in all text and tables and binominal nomenclature should be paid attention to (line 200). Do not start with a capital letter after a comma Line 202. References are generally very old. For the 2024 article, too many old references (2004,2012,2011,2014,2013) should be supplemented with new references.

Specific comments

There is no information on microbial load and sterilization in the procurement of fish in the material method.

Lines 125, 131 and 135 are not referenced. There is no information on why the acid levels, time and inputs selected in the marination procedure were chosen. In the analysis periods, for the marinades matured in 21 days, why the 0,3 and 15th day analyzes were made and on the basis of which parameter the marination was completed.

Although microbial analyzes are appropriate in the analysis, the lack of salinity and acidity analyzes despite sensory evaluation is a serious deficiency. Again, since it is an oily fish, PV and TBA are correct analyses, but since the goal in marinating is to reduce acidity and activate protease enzymes, it would be meaningful to have TVB-N data in the analysis.

In the discussion part, especially pH, lactic acid, coliform and total bacteria results should be discussed with a holistic approach. The role of bacteria dominating each other and the role of pH in this process should be explained.

Conclusion

A clear recommendation statement and process description should be made

Reviewer #2: - I recommend that the authors shorten the abstract.

- L55-56: Check the sentence: What is 4.3?

- Introduction: This section contains too much general information. I recommend shortening it. Also, provide at least 3 paragraphs of introduction instead of one.

- Hypothesis and objective: These parts in the introduction need to be improved and rewritten.

- Add a materials and methods section or an experimental section in the manuscript.

- L 120: What are the conditions for obtaining anchovy fillets?

- L125: Did the authors follow any reference method for preparation? If so, include it as well.

- L132: Why 5 min? What is the temperature of the inoculum and the samples? Report it.

- Marination process: There are many typos in this section, especially using symbols. Check it carefully. Similarly, check for typos throughout the manuscript.

- Provide a diagram or a flowchart for the sample preparation, inoculation, and marination.

- L143: Elaborate this section.

- L164: Trained panelists? Experts in the field of what? Or did you train them specifically for this study? Include details as well.

- L172: Why duplicate? Why not triplicate?

- Statistical analysis: This part is not clear. Please elaborate.

- Table 1: The columns that contain 0 values should use - or leave blank. If the authors use -, then provide a note below the table, as growth not found.

- Section Morganella Psychrotolerans: This section lacks discussion. Very few literature reviews are provided. Please elaborate it.

- I recommend splitting Table 1 into many, providing individual microorganism data and placing it below their discussion. For now, Table 1 is overloaded, not convenient to read, and overwhelming.

- A similar recommendation goes for Table 2 as well.

- Sensory analysis: Provide this part in figure format.

- Conclusion: This part needs to be concise and targeting main points.

- Overall: This manuscript contains important data, which is useful, but the authors have written too much when it is not necessary. I recommend that the authors check thoroughly the writing style and include only the necessary information.

6. PLOS authors have the option to publish the peer review history of their article (what does this mean?). If published, this will include your full peer review and any attached files.

Reviewer #1: No

Reviewer #2: No

---

## [Author Response · Author response to Decision Letter 0]

14 Mar 2024

Answer to Reviewers' comments

Dear Editor,

Our manuscript, entitled: “Effects of different marination conditions on the physico-chemical and microbiological quality of anchovy (Engraulis encrasicolus) fillets inoculated with Morganella psychrotolerans during cold storage” was revised carefully according to the comments from reviewers. All changes made in the text were track changes and a clean copy was also created. We would like to thank to the reviewers for their kind and useful comments for the contribution of paper.

Below you will find our point- by- point responses to the reviewers’ comments (Q: questions

R: response) 

We hope that the revised paper will meet the journal’s standard.

Thank you very much again for your attention to our paper．

Reviewer 1

Q: Too many spelling and grammatical errors in the text.

R: Thank you for your suggestions. Manuscript edited carefully and all spelling and gramatical errors were corrected.

Q: The title is not written with due care and the scientific names in the text are not written with due care.

R: Thank you for the comment. Title was edited carefully and scientific names were corrected through the manuscript.

Q: References are generally very old. For the 2024 article, too many old references (2004,2012,2011,2014,2013) should be supplemented with new references.

R: thank you very much for this valuable suggestion. References were revised and updated.

Q: Lines 125, 131 and 135 are not referenced.

R: Thank you for the carefull review. References were added.

Q: There is no information on why the acid levels, time and inputs selected in the marination procedure were chosen. In the analysis periods, for the marinades matured in 21 days, why the 0,3 and 15th day analyzes were made and on the basis of which parameter the marination was completed.

R: Thank you very much for this valuable comment. The marination parameters were chosen according to the pre-studies which were conducted before the main study. Different acetic acid and salt concentrations were applied to the anchovy fillets and evaluated in terms of sensorial parameters. Then the most selected concentrations were chosen. Additionally, in the literature mostly those concentrations are applied. Maturation time is also based on the sensory evaluation of pre-study. These informations were not included in the manuscript in order to avoid too much writing.

Q: Although microbial analyzes are appropriate in the analysis, the lack of salinity and acidity analyzes despite sensory evaluation is a serious deficiency. Again, since it is an oily fish, PV and TBA are correct analyses, but since the goal in marinating is to reduce acidity and activate protease enzymes, it would be meaningful to have TVB-N data in the analysis.

R: Thank you very much for valuable suggestion. Unfortunately our infrastructure were not suitable and the budget of the project were not enough for adding additional analyses like TVB-N.

Q: In the discussion part, especially pH, lactic acid, coliform and total bacteria results should be discussed with a holistic approach. The role of bacteria dominating each other and the role of pH in this process should be explained.

R: thank you for this suggestion. These sections were revised carefully and necessary information was given.

Reviewer 2

Q: I recommend that the authors shorten the abstract.

R: Thank you for the suggestion. Abstract was shortened and edited carefully.

Q: Check the sentence: What is 4.3?

R: Sentence was revised.

Q: Introduction: This section contains too much general information. I recommend shortening it. Also, provide at least 3 paragraphs of introduction instead of one.

- Hypothesis and objective: These parts in the introduction need to be improved and rewritten.

- Add a materials and methods section or an experimental section in the manuscript.

R: Thank you so much for this valuable comment. Introduction section was improved and rewritten. Additionally, suggested informations were given in this section.

Q: L 120: What are the conditions for obtaining anchovy fillets?

R: Thank you for this comment. Anchovy fillets were obtained from a local marked and details were included in the related section.

Q: L125: Did the authors follow any reference method for preparation? If so, include it as well.

R: Thank you very much for the suggestion. The preparation was conducted according to the German Collection of Microorganisms and Cell Culture (DSMZ, Braunschweig, Germany) specification as they suggested. Also reference was added.

Q: L132: Why 5 min? What is the temperature of the inoculum and the samples? Report it.

R: Thank you very much for the valuable question. The inocultaion procedure was conducted according to the method of Ucak et al. (2019) where they studies the same bacteria in marinated fish. Related reference were also added in this section.

Q: Marination process: There are many typos in this section, especially using symbols. Check it carefully. Similarly, check for typos throughout the manuscript.

R: This section was edited and suggested points were corrected.

Q: Provide a diagram or a flowchart for the sample preparation, inoculation, and marination.

R: Thank you for the valuable contribution. A flowchart was included in the material and method section.

Q: L143: Elaborate this section.

R: Section was elaborated and detailed.

Q: L164: Trained panelists? Experts in the field of what? Or did you train them specifically for this study? Include details as well.

R: Thank you for the valuable question. Details were given in the related section. The panelists were not trained but they were selected from people who have fish and marinated fish consumption habbit.

Q: L172: Why duplicate? Why not triplicate? Statistical analysis: This part is not clear. Please elaborate.

R: The section was edited. There was a mistake due to copy-paste from previous studies. All measurements were carried out in triplicate.

Q: Table 1: The columns that contain 0 values should use - or leave blank. If the authors use -, then provide a note below the table, as growth not found.

R: All tables were revised as suggested. We used ‘’-‘’ and put a note below the table as growth not found.

Q: Section Morganella Psychrotolerans: This section lacks discussion. Very few literature reviews are provided. Please elaborate it.

R: Thank you very much for the suggestion. This section was edited and more literature were added.

Q: I recommend splitting Table 1 into many, providing individual microorganism data and placing it below their discussion. For now, Table 1 is overloaded, not convenient to read, and overwhelming.

R: Thank you very much for the contribution. All tables were splitted and edited as recommended.

Q: A similar recommendation goes for Table 2 as well.

R: All tables were splitted and edited as recommended.

Q: Sensory analysis: Provide this part in figure format.

R: This part was rewritten and the table of this section converted to the figures.

Q: Conclusion: This part needs to be concise and targeting main points.

R: Thank you for the valuable comment. Conclusion section was edited and rewritten.

Q: Overall: This manuscript contains important data, which is useful, but the authors have written too much when it is not necessary. I recommend that the authors check thoroughly the writing style and include only the necessary information.

R: Thank you for your valuable time in commenting on the article. All comments have been responded and corrections have been made to the article.

---

## [Decision Letter · Decision Letter 1]

27 Mar 2024

Effects of different marination conditions on the physico-chemical and microbiological quality of anchovy (Engraulis encrasicolus) fillets inoculated with Morganella psychrotolerans during cold storage

PONE-D-23-44219R1

Dear Dr. Oluwatosin Abidemi Ogunkalu

We’re pleased to inform you that your manuscript has been judged scientifically suitable for publication and will be formally accepted for publication once it meets all outstanding technical requirements.

Kind regards,

Luca Nalbone, Ph.D.

Academic Editor

PLOS ONE

Additional Editor Comments (optional):

Reviewers' comments:

Reviewer's Responses to Questions

**Comments to the Author**

1. If the authors have adequately addressed your comments raised in a previous round of review and you feel that this manuscript is now acceptable for publication, you may indicate that here to bypass the “Comments to the Author” section, enter your conflict of interest statement in the “Confidential to Editor” section, and submit your "Accept" recommendation.

Reviewer #1: All comments have been addressed

Reviewer #2: All comments have been addressed

2. Is the manuscript technically sound, and do the data support the conclusions?

Reviewer #1: Yes

Reviewer #2: Yes

3. Has the statistical analysis been performed appropriately and rigorously? 

Reviewer #1: Yes

Reviewer #2: Yes

4. Have the authors made all data underlying the findings in their manuscript fully available?

Reviewer #1: Yes

Reviewer #2: Yes

5. Is the manuscript presented in an intelligible fashion and written in standard English?

Reviewer #1: Yes

Reviewer #2: Yes

6. Review Comments to the Author

Reviewer #1: It is an interesting document in the fields of processing and preservation of fish and products. The authors have addressed all my comments and improved manuscript

Reviewer #2: (No Response)

7. PLOS authors have the option to publish the peer review history of their article (what does this mean?). If published, this will include your full peer review and any attached files.

Reviewer #1: No

Reviewer #2: No

---

## [Editor Report · Acceptance letter]

26 Apr 2024

PONE-D-23-44219R1 

PLOS ONE

Dear Dr. Ogunkalu, 

I'm pleased to inform you that your manuscript has been deemed suitable for publication in PLOS ONE. Congratulations! Your manuscript is now being handed over to our production team.

Kind regards, 

on behalf of

Dr. Luca Nalbone 

Academic Editor

PLOS ONE